Analysis in silico of the functional interaction between WNT5A and YAP/TEAD signaling in cancer

Astudillo Pablo pablo.astudillo@uautonoma.cl
Instituto de Ciencias Biomédicas, Facultad de Ciencias de la Salud, Universidad Autónoma de Chile , Santiago , Chile
Verghese Shilpi
Electronic publication date: 2021 Feb 17
Publication date: 2021
Volume: 9
Electronic Location ID: e10869
Received 2020 Oct 2; Accepted 2021 Jan 10
Copyright: ©2021 Astudillo
Copyright year: 2021
Copyright holder: Astudillo
License: This is an open access article distributed under the terms of the Creative Commons Attribution License, which permits unrestricted use, distribution, reproduction and adaptation in any medium and for any purpose provided that it is properly attributed. For attribution, the original author(s), title, publication source (PeerJ) and either DOI or URL of the article must be cited.
License URL: https://creativecommons.org/licenses/by/4.0/

Keywords: Wnt5a, YAP, Cancer, Signaling pathways, Transcription factors

Funding: ANID PAI Convocatoria Nacional Subvención a la Instalación en la Academia, Convocatoria Año 2017 No. 77170063 This work was supported by the following grant: ‘ANID PAI Convocatoria Nacional Subvención a la Instalación en la Academia, Convocatoria Año 2017, No. 77170063’. The funders had no role in study design, data collection and analysis, decision to publish, or preparation of the manuscript.

==============================
To date, most data regarding the crosstalk between the Wnt signaling pathway and the YAP/TAZ transcriptional coactivators focuses on the Wnt/β-catenin branch of the pathway. In contrast, the relationship between the non-canonical Wnt pathway and YAP/TAZ remains significantly less explored. Wnt5a is usually regarded as a prototypical non-canonical Wnt ligand, and its expression has been related to cancer progression. On the other hand, YAP/TAZ transcriptional coactivators act in concert with TEAD transcription factors to control gene expression. Although one article has shown previously that WNT5A is a YAP/TEAD target gene, there is a need for further evidence supporting this regulatory relationship, because a possible YAP/Wnt5a regulatory circuit might have profound implications for cancer biology. This article analyzes publicly available ChIP-Seq, gene expression, and protein expression data to explore this relationship, and shows that WNT5A might be a YAP/TEAD target gene in several contexts. Moreover, Wnt5a and YAP expression are significantly correlated in specific cancer types, suggesting that the crosstalk between YAP/TAZ and the Wnt pathway is more intricate than previously thought.

Introduction

The Wnt signaling pathway modulates key processes during development and homeostasis, and alterations in this pathway have been related to disease (Logan & Nusse, 2004). This pathway is commonly divided into two main branches. The Wnt/β-catenin pathway, perhaps the best characterized, depends on the stabilization and nuclear translocation of β-catenin. Mechanistically, the activation of this pathway relies on the binding of Wnt ligands to Frizzled receptors and the co-receptors LRP5/6. The absence of Wnt stimulation leads to the degradation of β-catenin by a so-called β-catenin destruction complex, while activation of the pathway provokes the translocation of components of this destruction complex to the plasma membrane, triggering the formation of a complex known as the signalosome, which in turns becomes internalized (Bilic et al., 2007; Taelman et al., 2010). The internalization of the signalosome results in inactivation of the β-catenin destruction complex, and newly synthesized β-catenin can then translocate to the nucleus, binding to TCF/LEF transcription factors and controlling gene expression (for a review, see Nusse & Clevers, 2017). Importantly, abnormal Wnt/β-catenin signaling is commonly associated with cancer initiation (Anastas & Moon, 2012).

On the other hand, there is a second branch in the Wnt pathway, which comprises several pathways characterized by their independence from β-catenin stabilization (summarized in Semenov et al., 2007). Therefore, this pathway is commonly termed ‘non-canonical’ (or ‘β-catenin independent’) Wnt pathway. Wnt ligands bind to Frizzled receptors and various co-receptors, particularly Ror1/2, triggering intracellular events that lead to changes in cell behavior (Schlessinger, Hall & Tolwinski, 2009; Van Amerongen, 2012). Wnt ligands are usually classified as canonical or non-canonical, depending on their ability to activate specific Wnt branches (Kikuchi, Yamamoto & Kishida, 2007; Van Amerongen, Mikels & Nusse, 2008). For instance, Wnt3a is traditionally regarded as a canonical ligand, whereas Wnt5a is considered a non-canonical ligand. However, the specific pathway activated by each Wnt ligand might ultimately depend on the cellular context, such as combinations of Frizzled receptors and co-receptors (Niehrs, 2012).

The Wnt pathway can interact with other signaling pathways to control cell behavior. In this regard, data published in the last decade has highlighted the strong link between the Wnt pathway and the transcriptional coactivators YAP and TAZ (briefly discussed below). YAP and TAZ are mediators of the Hippo signaling pathway (Moya & Halder, 2019; Zheng & Pan, 2019; Dey, Varelas & Guan, 2020). These proteins have been extensively related to mechanotransduction, but this role might be independent of upstream components of the Hippo pathway (Dupont et al., 2011).

A remarkable observation in some cancer types is the stiffening of the tumor microenvironment (TME) due to extracellular matrix deposition and crosslinking (Kai, Laklai & Weaver, 2016; Kai, Drain & Weaver, 2019). As cancer cells invade the TME, they find a stiffened matrix, activating signaling pathways mediated by integrins and YAP/TAZ, among other proteins. When activated, YAP and TAZ translocate to the nucleus, where they bind to TEAD transcription factors to modulate gene expression (Stein et al., 2015; Zanconato et al., 2015). Therefore, a possible functional relationship between YAP, TAZ, and proteins involved in Wnt signaling might link the Wnt pathway to mechanosensing.

As noted above, many reports have shown that the Wnt pathway crosstalks with YAP and TAZ. YAP and TAZ functionally and physically interact with the Wnt machinery, including Dishevelled (Varelas et al., 2010), APC (Cai et al., 2015), β-catenin (Imajo et al., 2012; Azzolin et al., 2012), and others (Azzolin et al., 2014). Consequently, the literature has focused mainly on the relationship between the Wnt/β-catenin pathway and YAP/TAZ. This focus might be explained by the fact that, similar to the involvement of the Wnt/β-catenin pathway in cancer, YAP and TAZ have also been linked to this disease, particularly cancer initiation (Zanconato, Cordenonsi & Piccolo, 2016). However, the non-canonical Wnt pathway, particularly Wnt5a, also plays a crucial role in cancer (Kikuchi et al., 2012). Therefore, it is of great relevance to determine whether Wnt5a might also crosstalk with YAP and TAZ. To date, one report demonstrated that WNT5A is a YAP/TEAD target gene (Park et al., 2015) and that Wnt3a and Wnt5a can activate YAP/TAZ through an ‘alternative’ pathway independent of LRP5/6 but requiring Ror1/2, G α12∕13 and Rho, thus lending support to this putative crosstalk. In this article, publicly available ChIP-Seq, gene expression, and protein expression data are analyzed to further explore this possible functional relationship.

Materials & Methods

Analysis of the WNT5A TSS upstream sequence

The CistromeDB database allows querying for transcription factors (TF) likely binding to regions upstream the transcriptional start site (TSS) of a given gene (Mei et al., 2017; Zheng et al., 2019). A region ∼10 kb upstream of the TSS of the WNT5A gene was analyzed using the ‘ToolKit’ feature, querying for ‘Transcription factor, chromatin regulator,’ employing the NM_003392 transcript for analysis.

As a reference value to compare the TF binding sites (TFBS) found in the WNT5A gene, the CCN2 gene, encoding for the Connective Tissue Growth Factor (CTGF) protein, was analyzed using CistromeDB. CCN2 is an established TEAD target gene (Zhao et al., 2008). The top peaks found for either YAP1 or TEAD1/4 binding sites were retrieved. The values obtained are, respectively: YAP1, 0.659; TEAD1, 0.626; TEAD4, 0.757.

The ChIP-Atlas (Oki et al., 2018) integrates publicly available ChIP-seq data. The ‘Peak Browser’ feature was used to search for TFs or chromatin marks (H3K27ac, H3K4me1, and H3K4me3), selecting “All cell types” and a threshold for significance = 50. The data was mapped onto the IGV genome browser (Thorvaldsdottir, Robinson & Mesirov, 2013), spanning a region of ∼10 kb (chr3:55,520,700-55,530,700). The same region was analyzed with the ECR browser (Ovcharenko et al., 2004) to search for identity between the WNT5A gene and the corresponding gene in the species indicated in the main text, retrieving percent identity plots (PIPs) for evolutionary conserved regions (ECRs) and using the default parameters (minimum ECR identity = 70%; minimum ECR length = 100 pb). Finally, the KnockTF database (Feng et al., 2020) was queried to search for publicly available experiments correlating the knockdown of TEAD TFs with changes in WNT5A expression.

Gene and protein expression data

For analysis of gene expression, the web tools TIMER (version 2.0) (Li et al., 2017b) and GEPIA (version 2.0) (Tang et al., 2019) were used. In TIMER, the ‘Gene Correlation’ module was used. WNT5A was selected as the ‘interested’ gene, and the expression of YAP1, adjusted by tumor purity, was analyzed. A heatmap depicting partial Spearman rho values (degree of correlation between WNT5A and YAP1) was obtained. Of note, only tumor expression data is available for the cancer types with the highest correlation values. In GEPIA, a ‘Correlation Analysis’ was performed, using either single genes (WNT5A, YAP1) or gene signatures (CTGF plus ANKRD1). In each case, Spearman correlation coefficients were computed. Testicular Germ Cell Tumors (TGCT), PAAD (Pancreatic Adenocarcinoma), and BRCA (Breast Invasive Carcinoma) tumor types were evaluated. Since TCGA ‘Normal’ data was not available for TGCT, only GTEx tissue data was used for tumor/normal comparisons to maintain consistency across the analysis.

For protein expression data, the cBioPortal (Cerami et al., 2012; Gao et al., 2013) and The Cancer Proteome Atlas (TCPA) (Li et al., 2013; Li et al., 2017a) tools were used. In cBioPortal, one TGCT study (TCGA PanCancer Atlas) was queried for WNT5A and YAP1, using a z-score threshold =  ± 1.0 for both mRNA (relative to diploid samples, RNA Seq V2 RSEM) and protein (Reverse Phase protein Arrays, RPPA) expression (149 samples). The Spearman and Pearson correlation coefficients and their respective p-values were computed automatically by cBioPortal. In TCPA, the ‘Correlation Analysis’ option was used to query the TCGA datasets (TGCT, 118 samples; PAAD, 105 samples; BRCA, 901 samples). The graphs showing the correlation between YAP1 and phosphorylated JNK (pJNK) or AKT (pAKT) were downloaded, and the statistical information (Spearman correlation coefficients) was retrieved from the respective data tables.

Results

To corroborate whether WNT5A expression is modulated by YAP, publicly available ChIP-Seq data was first explored, using CistromeDB. A region spanning ∼10 kb upstream the transcription start site (TSS) of the WNT5A gene was analyzed. CistromeDB reveals a subset of TFs with high regulatory potential (RP) score (Fig. 1A). Notably, YAP1 was among the hits retrieved by the analysis, with two binding sites with RP values ranging from 0.48 to 0.67 (Fig. 1A). These values are similar to those found in the well-established YAP/TEAD target gene CCN2/CTGF (see ‘Materials & Methods’). CistromeDB also reveals binding sites for TEAD1 and TEAD4, albeit with low RP scores (the top RP scores were 0.39 for TEAD1, and 0.28 for TEAD4). LATS2 has also been shown to be a YAP/TEAD target gene (Moroishi et al., 2015; Molina-Castro et al., 2020). Analysis of the LATS2 gene in CistromeDB also confirms binding sites for TEAD1, with RP scores ranging from 0.3 to 0.64, similar to the values observed for WNT5A (Fig. 1B). Interestingly, the analysis using CistromeDB also reveals YAP binding sites, with a high RP value, for the YAP1 gene (Fig. 1C), suggesting that YAP1 might self-regulate its expression through TEAD-mediated transcription. Studies reporting ChIP-Seq analysis have shown that the YAP1 gene might be a YAP/TEAD target gene (for instance, see supplementary information in Stein et al., 2015; Zanconato et al., 2015).

Figure 1 Binding sites with high regulatory potential scores for WNT5A, LATS2, and YAP1.

Data was obtained using CistromeDB, spanning a region of ∼10 kb upstream the transcription start site (TSS) of the WNT5A (A), LATS2 (B), and YAP1 (C) genes. YAP/TEAD proteins are highlighted in bold.

To further corroborate the analysis presented above, publicly available ChIP-Seq data was analyzed using the ChIP-Atlas. This analysis reveals two regions with potential binding sites for YAP1, as well as for TEAD1 and TEAD4, upstream of the WNT5A gene (Fig. 2A). It has been reported that YAP/TAZ and AP-1 TFs co-occupy the same genomic regions in a majority of TEAD target genes (Zanconato et al., 2015). Interestingly, the AP-1 transcription factors JUN and FOS also overlap with these YAP1 and TEAD1/4 potential binding sites (Fig. 2A), defining two distinctive potential regulatory regions.

Figure 2 YAP/TEAD binding sites upstream the WNT5A transcription start site (TSS).

(A) ChIP-Seq data mapped onto a region spanning 10 kb upstream of the TSS of the WNT5A gene, using the IGV browser. The data for YAP1, transcription factors (TEAD1, TEAD4, JUN, FOS), and histone modifications (H3K27ac, H3K4me1, and H3K4me3), correspond to separate experiments, represented by different colors. See the main text for details. (B) Evolutionary conserved regions (ECRs) between the human, mouse, and rat Wnt5a gene. The same region shown in (A) was analyzed using the ECR Browser. The dashed box shows the matching region enriched for YAP/TEAD binding, according to the ChIP-Seq data. (C) Data from the specified KnockTF dataset showing WNT5A (in bold) as a downregulated gene after TEAD4 knockdown.

Figure 3 Correlation between WNT5A and YAP1 expression across cancer types.

(A) Heat map (Spearman correlation coefficients) retrieved from TIMER, showing the correlation between WNT5A and YAP1 expression across the indicated cancer types (ACC, Adrenocortical carcinoma; BLCA, Bladder Urothelial Carcinoma; BRCA, Breast invasive carcinoma; CESC, Cervical squamous cell carcinoma and endocervical adenocarcinoma; CHOL, Cholangiocarcinoma; COAD, Colon adenocarcinoma; DLBC, Lymphoid Neoplasm Diffuse Large B-cell Lymphoma; ESCA, Esophageal carcinoma; GBM, Glioblastoma multiforme; HNSC, Head and Neck squamous cell carcinoma; KICH, Kidney Chromophobe; KIRC, Kidney renal clear cell carcinoma; KIRP, Kidney renal papillary cell carcinoma; LGG, Brain Lower Grade Glioma; LIHC, Liver hepatocellular carcinoma; LUAD, Lung adenocarcinoma; LUSC, Lung squamous cell carcinoma; MESO, Mesothelioma; OV, Ovarian serous cystadenocarcinoma; PAAD, Pancreatic adenocarcinoma; PCPG, Pheochromocytoma and Paraganglioma; PRAD, Prostate adenocarcinoma; READ, Rectum adenocarcinoma; SARC, Sarcoma; SKCM, Skin Cutaneous Melanoma; STAD, Stomach adenocarcinoma; TGCT, Testicular Germ Cell Tumors; THCA, Thyroid carcinoma; THYM, Thymoma; UCEC, Uterine Corpus Endometrial Carcinoma; UCS, Uterine Carcinosarcoma; UVM, Uveal Melanoma). (B–G) Correlation between WNT5A and YAP1 expression in the indicated cancer types, according to data from GEPIA, using tumor (B–D) and GTEx normal (E–G) data. Spearman correlation coefficients were automatically computed and are shown at the top of each plot. TPM, transcript per million.

Acetylation of histone H3 at lysine 27 (H3K27ac) as a signature of active enhancers is positively correlated with TEAD binding (Stein et al., 2015), while Zanconato and colleagues used the presence of H3K4me1 peaks not overlapping with H3K4me3 to identify enhancers (Zanconato et al., 2015). Mapping H3K27ac, H3K4me1, and H3K4me3 data to the upstream genomic region of WNT5A on the IGV browser reveal partial overlap with YAP1, TEAD1, TEAD4, JUN, and FOS (Fig. 2A). H3K27ac extensively marked the region under analysis. Of note, one of these regions is well-conserved in mice and rats (Fig. 2B, dashed box). Finally, using the KnockTF database to search for genes down-regulated after knockdown of TEAD TFs indicates that WNT5A is among the most down-regulated genes after TEAD4 knockdown in the SNU216 gastric cell line (Fig. 2C). It is worth noting that this data comes from an independent study (Lim et al., 2014), published before the article from Park and colleagues (Park et al., 2015).

In summary, in silico analysis corroborates that WNT5A is a TEAD target gene. Moreover, these results indicate that YAP might modulate WNT5A expression in different contexts and suggest the existence of a regulatory circuit comprising Wnt5a, YAP, and possibly LATS2. Given that both Wnt5a and abnormal YAP/TAZ signaling have been linked to cancer (see ‘Introduction’), this relationship was further explored using cancer gene expression data.

Analysis of gene expression data using TIMER 2.0 shows a moderate-to-strong correlation between WNT5A and YAP1 in several cancer types (Fig. 3A). Interestingly, the strongest correlation was observed for Testicular Germ Cell Tumors (TGCT; Spearman correlation = 0.7), a cancer type with little information about the involvement of the Wnt pathway (see ‘Discussion’). A moderate (Spearman correlation value > 0.5) correlation was also observed for PAAD (Pancreatic Adenocarcinoma), THYM (Thymoma), UVM (Uveal Melanoma), UCS (Uterine Carcinosarcoma), SKCM-Primary (Skin Cutaneous Melanoma), BRCA-LumA (Breast Invasive Carcinoma) and DLBC (Diffuse Large B-Cell Lymphoma) (Fig. 3A). This correlation was corroborated for TGCT, BRCA and PAAD (Figs. 3B–3D). Importantly, there was a lower correlation between WNT5A and YAP1 in normal tissues for testis, breast, and pancreas (Figs. 3E–3G).

Wnt5a has been reported to either activate or inhibit Wnt/β-catenin signaling in several cellular contexts (Torres et al., 1996; He et al., 1997; Ishitani et al., 1999; Ishitani et al., 2003; Topol et al., 2003; Westfall et al., 2003; Mikels & Nusse, 2006). By modulating the Wnt/β-catenin pathway, Wnt5a might release YAP from the β-catenin destruction complex, allowing YAP to promote the expression of TEAD target genes. Alternatively, Wnt5a might promote the assembly of the β-catenin destruction complex, thus sequestering YAP and TAZ, concomitantly abolishing the expression of TEAD target genes.

In consequence, it was evaluated whether WNT5A levels correlated with TEAD target gene expression in the cancer types where a moderate to strong correlation was observed, using GEPIA. For TGCT, a strong correlation (Spearman correlation = 0.69) between WNT5A and a gene signature composed of CTGF and ANKRD1 (two well-established TEAD target genes) was observed. In contrast, only weak correlations were observed for PAAD and BRCA cancer (Fig. 4). Notably, when comparing cancer and normal (GTEx data) expression, this correlation was greatly lost in both testis and pancreas (Fig. 4). In addition, the analysis of gene and protein expression data in TGCT showed that high WNT5A levels correlated with high YAP protein expression (Fig. 5A). Therefore, these results suggest that, at least in TGCT, Wnt5a might promote YAP/TAZ stability and activity. In addition, the correlation between WNT5A and the CTGF/ANKRD1 signature as a readout for YAP/TEAD activity might be correlated with cancer progression in both TGCT and PAAD.

The activation of the Wnt/β-catenin signaling pathway leads to specific and well-established readouts, such as phosphorylation of LRP6 and nuclear accumulation of β-catenin. On the contrary, signaling mediated by non-canonical Wnt ligands activates less-specific readouts, such as JNK or AKT phosphorylation. The TCGA contains protein (RPPA) expression data for approximately 300 proteins, thus allowing to perform preliminary exploration of the activation of certain signaling pathways. Data from the TCPA was explored for TGCT, BRCA, and PAAD cancer. Interestingly, both JNK (pT183, Y185) and AKT (pT308) phosphorylation were highly correlated with YAP expression in TGCT cancer (Fig. 5B), but not in BRCA (Spearman’s rank correlation coefficient, pJNK = 0.12713; pAKT = 0.19721; p-values, pJNK = 0.00016438; pAKT = 4.1085e−9) or PAAD (Spearman’s rank correlation coefficient, pJNK = −0.045635; pAKT = −0.0853; p-values, not significant). These results must be interpreted with caution since high Wnt5a expression might lead to activation of other intracellular effectors, such as Rho GTPases, in these cancer types (Schlessinger, Hall & Tolwinski, 2009).

Figure 4 Correlation between WNT5A expression and a YAP/TEAD gene signature.

Tumor (A–C) and the corresponding GTEx normal tissue (D–F) data are shown for the indicated cancer types, and the correlation between WNT5A and the ‘YAP/TEAD’ signature (CTGF plus ANKRD1) was estimated using the ‘Gene Signature’ option in GEPIA. Spearman correlation coefficients were automatically computed and are shown at the top of each plot.

Figure 5 Correlation between WNT5A and YAP protein expression and JNK/AKT phosphorylation in Testicular Germ Cell Tumors.

(A) Correlation between WNT5A and YAP protein expression in TGCT cancer. The data was retrieved from cBioPortal (see ‘Methods’ for the parameters used for the analysis). (B) Correlation between YAP protein expression and JNK (left) and AKT (right) protein phosphorylation, according to The Cancer Proteome Atlas.

Collectively, the analysis presented in this article suggests that WNT5A expression is modulated by YAP/TEAD in several cancer types, although likely leading to different cellular outcomes depending on the tissue or cell type. More importantly, WNT5A correlates with a YAP/TEAD signature and pJNK/pAKT in testicular germ cell cancer, thus providing potentially novel evidence for a role of the non-canonical Wnt pathway in this cancer type.

Discussion

To date, most articles assessing the relationship between YAP, TAZ, and the Wnt signaling pathway have focused on the involvement of YAP/TAZ in the Wnt/β-catenin signaling branch. Research from several groups has shown that YAP/TAZ interacts with the β-catenin destruction complex and other Wnt components, as noted above. According to the prevailing model, activation of the Wnt/β-catenin pathway leads to the disassembly of the β-catenin destruction complex, thus releasing YAP and TAZ. In turn, YAP and TAZ dynamically shuttle across the nucleus, to either co-repress or co-activate the transcription of target genes by interacting with TEAD transcription factors (Manning, Kroeger & Harvey, 2020).

In contrast, the possible interaction between the non-canonical Wnt pathway and YAP/TAZ remains less understood. However, growing evidence suggests that YAP and Wnt5a might functionally interact in cancer (Tu et al., 2019; Luo et al., 2020) and chronic kidney disease (Feng et al., 2018). Interestingly, chronic kidney disease is characterized by increased fibrosis; similarly, cancer progression is also accompanied by an increased stiffening of the TME, due to higher extracellular matrix deposition and collagen crosslinking. This suggests that Wnt5a and YAP/TAZ might functionally interact in certain cellular contexts. However, despite this growing body of evidence, there is a lack of deep mechanistic understanding of how Wnt5a and YAP might interact. Only one study addressed this interaction, showing that WNT5A is a TEAD target gene (Park et al., 2015).

Given the relevance of such a relationship, it is desirable to obtain further evidence supporting it. The availability of transcriptomic and proteomic data across cancer types allows surveying for potential correlations between gene signatures. In addition, the growing abundance of ChIP-Seq data facilitates the inquiry of potential regulatory relationships across cell types. This article analyzed publicly available data to corroborate whether YAP might regulate Wnt5a. In agreement with the previously existing observation reported by Park and colleagues (Park et al., 2015), ChIP-Seq data shows YAP/TEAD binding in some cancer cell lines (MSTO-211H, H2052, SF268, and SK-N-SH), according to the datasets included in the ChIP-Seq atlas. It will be interesting to determine whether YAP/TEAD binding to the WNT5A regulatory region is a feature of cancer cell lines or if this merely represents a need for additional studies in other cell types.

The data presented here show that an upstream region of the WNT5A gene contains two putative regulatory regions that might be responsible for this regulation. Given that previous reports showed a role for AP-1 TFs in the context of TEAD-dependent gene expression, the analysis presented in this article also employed AP-1 ChIP-Seq data, showing overlap with TEAD binding sites, strongly suggesting that these regions upstream the WNT5A TSS might constitute enhancers. It must be noted, however, that other TFs are likely needed to fine-tune the expression of WNT5A, thus explaining the results observed by surveying cancer expression data. Also, the present article does not address other genomic regions, which might also contain regulatory sequences. However, it has been reported that YAP/TAZ/TEAD complexes mostly modulate distant enhancers located beyond 1–2 kb upstream of the TSS (Stein et al., 2015; Zanconato et al., 2015). In addition, the WNT5A gene encodes two isoforms, Wnt5a-S (short) and Wnt5a-L (long) (Bauer et al., 2013). Both Wnt5a-S and Wnt5a-L have similar properties, but they differ in functions, including in some cancer types (Bauer et al., 2013; Huang et al., 2017). It will be interesting to determine whether TEAD-mediated modulation intersects with other signaling cues that might modulate the differential expression of these isoforms.

Notwithstanding, gene expression data showed that WNT5A and YAP1 expression is moderately-to-strongly correlated in several cancer types. This article focused on three cancer types: breast (BRCA) and pancreatic (PAAD) cancer, due to their clinical relevance; and testicular germ cell (TGCT) cancer, since little is known about the role of the Wnt pathway in this cancer type. For BRCA and PAAD, the analysis presented here showed that, although there is a moderate correlation between WNT5A and YAP1, this does not translate into an increased TEAD signature or JNK/AKT phosphorylation, commonly employed as a readout for non-canonical Wnt signaling. Further research will be needed to determine whether the WNT5A and YAP1 correlation has any functional relevance. In this regard, it must be noted that Wnt5a might exert different signaling outcomes. For instance, Wnt5a might modulate integrin adhesion dynamics (Kurayoshi et al., 2006; Matsumoto et al., 2010), while YAP itself also controls the expression of integrin adhesion components (Nardone et al., 2017). Both BRCA and PAAD are characterized by stiffening of the TME, and thus future research should assess whether Wnt5a and YAP might be involved in this fibrotic response. As noted above, recent evidence also suggests a functional relationship between YAP and Wnt5a in skin melanoma (Luo et al., 2020) and pancreatic adenocarcinoma (Tu et al., 2019), two cancer types where a moderate correlation between YAP1 and WNT5A expression is observed (Fig. 3). In the first case, the YAP/Wnt5a signaling axis is likely involved in modulating cell migration, while in the latter, it might be related to tumor growth. The precise molecular mechanisms in both cases remain to be fully elucidated, but this evidence helps to highlight the possible roles that might be regulated by the YAP/Wnt5a module.

The data presented in this article also shows that WNT5A levels are correlated with YAP protein levels and a TEAD signature (CTGF and ANKRD1 expression). Notably, the correlation between WNT5A and the TEAD signature is significantly lost in testis (GTEx) normal data, thus suggesting a potential relevance of this correlation. Moreover, YAP protein expression correlates with pJNK and pAKT. To date, the evidence regarding testicular germ cell cancer and Wnt signaling is limited to the Wnt/β-catenin pathway, although there is no clear role for this pathway in TGCT progression. Analysis of β-catenin expression and localization suggest that the Wnt/β-catenin pathway might be essential for normal spermatogenesis (Young et al., 2020); however, nuclear β-catenin is not seen in neoplastic germ cells (Chovanec et al., 2018; Young et al., 2020). In contrast, in vitro assays using a seminoma-derived cell line, TCam-2, suggests a possible role of the Wnt/β-catenin in cell viability and migration (Young et al., 2020). However, the abovementioned study did not address the role of Wnt5a, and the absence of mechanical and other biochemical signals from the tumor microenvironment might induce different responses in vivo.

On the other hand, comprehensive studies have shown that only a limited number of genes might be correlated with TGCT establishment (Shen et al., 2018). Therefore, the precise role of the Wnt pathway and YAP/TAZ in this cancer type remains to be fully established. Since TGCT are morphologically heterogeneous, and only a few genes have been conclusively correlated with TGCT establishment, perturbations in signaling pathways, rather than driving mutations, might likely play a crucial role in this disease. Therefore, the results relative to TGCT presented in this article might be of special interest.

Finally, it must be stressed that this article presents some limitations. The data obtained here must be properly explored both in vitro and in vivo. In this regard, the reports cited above offer possible models to explore a mechanistic relationship between YAP and Wnt5a. Secondly, the analysis presented in this article might be further expanded to include a Hippo signature (Wang et al., 2018). In addition, protein expression data related to the Wnt/β-catenin pathway (such as GSK-3β phosphorylation) must also be examined.

Conclusions

Altogether, the observations presented here suggest the further need to study the functional relationship between YAP/TAZ and the non-canonical Wnt pathway, particularly in the context of cancer. In addition, the crosstalk between YAP/TAZ and the Wnt pathway should be assessed from a systems biology perspective, considering possible feedback mechanisms and regulatory circuits, which might be perturbed in the context of cancer and other diseases.

Additional Information and Declarations

Competing Interests

Author Contributions

Data Availability

The author declares there are no competing interests.

Pablo Astudillo conceived and designed the experiments, performed the experiments, analyzed the data, prepared figures and/or tables, authored or reviewed drafts of the paper, and approved the final draft.

The following information was supplied regarding data availability:

All data is available at:

(a) CistromeDB database Toolkit (http://dbtoolkit.cistrome.org). Search terms (gene names) WNT5A, LATS2, YAP1, and CCN2 were explored using the ‘Result in Figure’ option. ‘Dynamic Plots’ were used to search for transcription factor binding sites, and the data was manually retrieved. ‘Static Plots’ were used in Fig. 1.

(b) ChIP-Seq Atlas (http://chip-atlas.org/peak_browser). Search terms were YAP1, TEAD1, TEAD4, JUN, FOS, H3K27ac, H3K4me1, and H3K4me3. Each search term was selected in the ‘Antigen’ menu, and the option ‘View on IGV’ was selected.

(c) ECR Browser (https://ecrbrowser.dcode.org); search term, chr3:55,520,700-55,530,700.

(d) GEPIA (version 2.0; http://gepia2.cancer-pku.cn/#correlation). Comparisons: a) WNT5A (‘Gene A’) versus YAP1 (‘Gene B’) (Fig. 3); b) WNT5A (‘Gene A’) versus CTGF and ANKRD1 (as ‘Gene Set B’) (Fig. 4). Cancer types: TGCT; PAAD; BRCA. Normal tissues (GTEx): Testis; Breast; Pancreas.

(e) TIMER (version 2.0; http://timer.cistrome.org). The ‘Gene_Corr’ function (in the ‘Cancer Exploration’ section) was used to compare WNT5A (‘Interested Gene’) and YAP1 (‘Gene Expression’).

(f) cBioPortal (https://www.cbioportal.org); TCGA PanCancer Atlas study; search terms, WNT5A; YAP1.

(g) The Cancer Proteome Atlas (https://tcpaportal.org/tcpa/correlation_analysis.html). The “TCGA Testicular Germ Cell Tumors (TGCT)” dataset (118 samples) was selected, and the following terms were queried using the ‘Search’ box: YAP1; JNK_pT183Y185; AKT_pT308. The plots were retrieved by first selecting the ‘Plot’ option, and the image was downloaded from the drop-down menu located in the upper right corner of the respective plot.

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
