# Peer review of "Analysis in silico of the functional interaction between WNT5A and YAP/TEAD signaling in cancer"

_PeerJ, doi:10.7717/peerj.10869_

## Round 0.1 · original submission · Minor Revisions

Please address the minor revisions addressed by the reviewers.

Reviewer 1 ·

Basic reporting

The manuscript discuss an important topic on Hippo pathway and Noncanonical Wnt regulation. The manuscript nicely shows the correlation between YAP-TEAD and Wnt5a expression across many cancer types and Wnt5a as potential TEAD target gene.

Experimental design

The experimental design is well suitable for publication.

Validity of the findings

The manuscript well-addresses previous findings and limitations and strongly suggest that Hippo pathway is closely related to noncanonical Wnt pathway by directly expressing Wnt5a ligand.

Additional comments

The manuscript discuss an important topic on Hippo pathway and Noncanonical Wnt regulation. The manuscript nicely shows the correlation between YAP-TEAD and Wnt5a expression across many cancer types and Wnt5a as potential TEAD target gene. The manuscript well-addresses previous findings and limitations and strongly suggest that Hippo pathway is closely related to noncanonical Wnt pathway by directly expressing Wnt5a ligand.

Reviewer 2 ·

Basic reporting

The article includes sufficient introduction and background to demonstrate how the work fits into the broader field of knowledge. Relevant prior literature was appropriately referenced.

Self-contained with relevant results to hypotheses.

Experimental design

Research question well defined, relevant & meaningful. It is stated how research fills an identified knowledge gap.

Validity of the findings

All underlying data have been provided; they are robust, statistically sound, & controlled.

Additional comments

I would be very glad to re-review the paper. In general, there is a lot of explanation of replicates and statistical methods used in the study. Also, there are enough explanations of the rationale for the study design. the submission is worthy of publication.

Reviewer 3 ·

Basic reporting

1.1 Overall the English is fluent in logic but the author needs to further polish the English grammar. For example, “on the other hand” was repeatedly used as conjunction phrase in line 293 and line 307.
1.2 The TME mentioned in line 86/87 is an emerging hot topic in cancer biology but this paragraph is confusing. The author proposed that YAP/TAZ may be independent of Hippo then guided the readers to TME and mentioned the activation of YAP/TAZ with TEAD binding. But TEAD binding is a canonical TF for Hippo. The author should further re-organize this paragraph to clarify the significance.
1.3 Regarding the crosstalk between Wnt5a and YAP/TAZ or Hippo, the author did not mention the activation of YAP/TAZ by non-canonical Wnt signaling, instead all the efforts in this version were paid to the activation of non-canonical Wnt by YAP/TAZ. Just as the author claimed in the abstract, “crosstalk between YAP/TAZ and the Wnt pathway is more intricate than previously thought”, those intercalating feedbacks should be stated in the introduction section. For example, the 2015 Cell paper cited in this manuscript (Park et al., 2015), YAP/TAZ was indeed activated by Wnt5a and even Wnt3a (through a non-canonical way), although the authors in the 2015 paper also demonstrated Wnt5a was a potential YAP/TEAD target gene.

Experimental design

no comment

Validity of the findings

3.1, please specify what RP score can be termed as high. Line 160.
3.2. In Figure 1A, no TEAD binding was found in Wnt5a promoter but in Figure 2A, there are significant occupancy of TEAD1/2 and YAP1. Will the author interpret this as a cancer type specific context for TEAD binding? What specific cancer type is it in Figure 2A ChIP seq datasets? Although there could be datasets variation, the author could provide more details here.
3.3, the author did find the YAP/TEAD-modulation of WNT5A in certain types of cancer. Further characterization of the types in the discussion will make this article more interesting to the broad field.

Additional comments

The overall design and rationale is important for further understanding of crosstalk between Wnt5a and YAP/TAZ. It would be even better if the author can suggest some underlying mechanisms for the cancer type (context) specific modulation between Wnt5a and YAP.

Reviewer 4 ·

Basic reporting

NA

Experimental design

NA

Validity of the findings

The authors very methodically show the presence of binding sites is conserved and is present on YAP1 (for both WNT5A and itself). Did the authors check this with certain negative regulators? How specific is the domain for this binding?
In the chip-seq data used to show the presence of binding sites, what kind of cell lines/samples is the data coming from?
In another KnockTF database, did the authors check for all down regulated/upregulated genes network to visualize any possible indirect relation between TEAD->WNT5a?
With the WNT5A and YAP1 correlation in different cancer- can the authors shed more lights on the cancer types with significant correlation? Is this a known phenomenon where WNT5A pathway is known to play an important role in the cancer progression? Can the authors label the GTEx (fig3b with tissue type, sample number etc)

Additional comments

The authors in this manuscript presented a completely in-silico and interesting method to show functional interaction of WNT5A and YAP/TEAD signaling in cancer. In their analysis they have used many bioinformatic approaches, web tools and public dataset like gene expression, chip-seq. Although Park et al., 2015 had addressed this interaction, this approach is quite applicable for other methods as well. It is a very well written manuscript with comprehensively citing literature.

Reviewer 5 ·

Basic reporting

In this study, the author, through in silico approaches, makes the assumption that Wnt5A is TEAD target and also about the role of YAP/TAZ signaling in regulation of non-canonical Wnt. The findings are the result of careful analysis of published data and the methodology is sound. The results will be of use for researchers particularly working in these areas.
Is there a role for Wnt5a-Yap/TAZ axis in other biological processes than cancer, for e.g., growth and development?
Did the author look at miRNA data to see if the regulation of Wnt5a by these Factors are dependent on those?
Wnt5a is known to have two isoforms. Comment on the possible differential effect on regulation of these.
In prostate cancers, promoter methylation of Wnt5a has been reported. Is there a negative correlation in the YAP/TAZ Chip data in these?

Experimental design

no comment

Validity of the findings

Is there a role for Wnt5a-Yap/TAZ axis in other biological processes than cancer, for e.g., growth and development?
Did the author look at miRNA data to see if the regulation of Wnt5a by these Factors are dependent on those?
Wnt5a is known to have two isoforms. Comment on the possible differential effect on regulation of these.
In prostate cancers, promoter methylation of Wnt5a has been reported. Is there a negative correlation in the YAP/TAZ ChIP-data in these?

---

## Round 0.2 · accepted · Accept

Congratulations. Thanks for addressing all concerns by the reviewers.

Reviewer 3 ·

Basic reporting

Clear and sound.

Experimental design

Good. No further comments.

Validity of the findings

Conclusions are well stated.

Additional comments

I think the author address the questions and comments raised by reviewers quite well. The revised version is suitable for publication.

Reviewer 4 ·

Basic reporting

NA

Experimental design

NA

Validity of the findings

NA

Additional comments

All the points have been addressed.

Reviewer 5 ·

Basic reporting

The author has responded adequately to all the queries. No further modifications are necessary.

Experimental design

No comment

Validity of the findings

No comment